# Electrophoretic Deposition and Characterization of Functional Coatings Based on an Antibacterial Gallium (III)-Chitosan Complex

**Muhammad Asim Akhtar [1,†], Zoya Hadzhieva [1,†], Ivo Dlouhý [2]** and **Aldo R. Boccaccini [1,*]**

1. Institute of Biomaterials, Department of Materials Science and Engineering,
   University of Erlangen-Nuremberg, 91058 Erlangen, Germany; asim.akhtar@fau.de (M.A.A.);
   zoya.hadzhieva@fau.de (Z.H.)
2. Institute of Physics of Materials ASCR, CEITEC IPM, Žižkova 22, 61662 Brno, Czech Republic;
   idlouhy@ipm.cz
* Correspondence: aldo.boccaccini@ww.uni-erlangen.de
† These authors contributed equally to this work and share first co-authorship.

**Abstract:** Despite their broad biomedical applications in orthopedics and dentistry, metallic implants are still associated with failures due to their lack of surface biofunctionality, leading to prosthesis-related microbial infections. In order to address this issue, the current study focuses on the fabrication and characterization of a novel type of antibacterial coating based on gallium (III)-chitosan (Ga (III)-CS) complex layers deposited on metallic substrates via electrophoretic deposition (EPD). Aiming for the production of homogeneous and monophasic coatings, a two step-procedure was applied: the first step involved the synthesis of the Ga (III)-CS complex, followed by EPD from suitable solutions in an acetic acid–aqueous solvent. The influence of Ga (III) concentration on the stability of the suspensions was evaluated in terms of zeta potential. Fourier transform infrared (FTIR) and energy dispersive X-ray (EDX) spectroscopic analyses indicated the chelation of CS with Ga (III) within the coatings, while scanning electron microscopy (SEM) confirmed that no additional metallic gallium deposited during EPD. Furthermore, the results demonstrated that the wettability, mechanical properties, swelling ability, and enzymatic degradation of the coatings were affected by the quantity of Ga (III) ions. Colony forming unit (CFU) tests showed a strong synergistic effect between CS and Ga (III) in inhibiting *Escherichia coli* strain growth compared to control CS samples. An in vitro study with MG-63 cells showed that Ga (III)-containing coatings were not toxic after 24 h of incubation.

**Keywords:** gallium; chitosan; electrophoretic deposition; antibacterial coatings; bioactivity

## 1. Introduction

Metallic biomaterials are widely utilized in the fields of orthopedics, dentistry, and bone reconstruction to provide internal support to biological tissues, due to their load-bearing ability and excellent mechanical properties [1–3]. Still, the long-term survivability of metallic devices is considerably restricted by their proneness to infections [4]. Bacterial surface contamination is a major concern because it could be associated with eventual hospitalization costs, morbidity, and a tendency for serious relapses [5]. The rate of infection is increasing, meaning infections are becoming difficult to treat due to increasing resistance of bacteria to antibiotics. With the passage of time, new antibiotic resistance mechanisms are emerging [6]. In addition, it has been seen in prosthetic joint infection therapy that infections due to Gram-negative bacteria are more unfavorable than those caused by

Gram-positive bacteria [7–9]. To address this challenge in biomedical implants, effective antibiotic-free antibacterial coatings are a possible solution.

One particular strategy in this regard is the functionalization of the metal prosthesis surface with biofunctional polymer-based coatings. Such coatings should provide bioactivity and antibacterial functionality to metallic implants. Chitosan (CS) has attracted the attention of researchers for developing biomedical coatings due to its wide range of appealing characteristics, including cytocompatibility, biodegradability, antifungal activity, and antibacterial properties [10,11]. Another key feature of CS is its ability to chelate with metal ions, especially those of transition elements [12]. This adsorption capacity of CS for metals is explained by its bifunctional nature, having active amine ($NH^{3+}$) and highly hydrophilic hydroxyl ($OH^-$) functional groups, which are accountable for donating their lone pair of electrons to the metal cation [12,13]. The incorporation of metal ions within a CS matrix is intriguing from the point of view of medical applications because it brings some very valuable properties to the final complex. First, such CS-metal ion complexes are reported to exert superior in vitro antibacterial activities in comparison to free CS or antimicrobial metal salts [14]. Another benefit of the complexation process is that it can be used to tailor the physical properties of the CS-metal derivatives and to ensure the controlled release of biologically active ions from the polymer matrix as a function of time [15]. Finally, the chelation of CS with therapeutic metal ions, which are able to interact with a number of biological entities and metabolic systems, offers an effective alternative to the conventional addition of antibiotics or growth factors [16]. Among different therapeutic metal ions, Ga(III) exerts inhibitory activity against numerous bacteria, such as *Staphylococcus aureus*, *Escherichia coli, Rhodococcus equi*, *Pseudomonas aeruginosa*, and *Acinetobacter baumannii* [17,18]. In addition to its efficiency against infectious disease, the salt gallium nitrate shows osteogenic [19,20] and antiresorptive [21] properties, as well as therapeutic activity against diverse types of cancer, including non-Hodgkin's lymphoma, bladder cancer, and malignancy-associated hypercalcemia [22,23]. These properties warrant the application of gallium ions in drug-releasing tissue engineering scaffolds, polymer-doped coatings on metallic implants, or various drugs with potentially favorable clinical outcomes [23–25]. In addition, the trivalent gallium ion, being classified as a hard acid, tends to form chelates with strong Lewis bases, in particular with both oxygen and nitrogen atoms from hydroxyl and amino groups on ligands, confirming its suitability to readily bind CS [26].

Over the years, a wide variety of techniques have been adopted to coat metallic substrates, including dip coating, thermal spraying, sol–gel, and layer-by-layer (LbL) deposition [27–29]. With these methods, different ceramic, metallic, and polymer materials can be deposited in order to improve the mechanical, tribological, or biological properties of the coatings. Among the various coating techniques, electrophoretic deposition (EPD) has been exploited for the preparation of CS-based composite coatings due to its versatility, cost-effectiveness, feasibility for room temperature processing, and possibility to control the coating properties and structure [30–32]. A simultaneous cationic co-deposition of CS and positively charged metal ions, including Ga [25], Mg [33], Sr [34], Ag [35], and Cu [36], has been previously investigated. However, in all mentioned studies a single-step EPD was applied. According to this method, the metal salt is dispersed in a polymer-based suspension directly before deposition so that the metal ions are only physically bonded or entrapped into the CS macromolecules. As a consequence, the available electrons on the cathode reduce the ions to nanoparticles through a combined process of in situ electrosynthesis and electrophoresis. It has been proven that the addition of salts in EPD suspensions affects the deposition mechanism of CS, which can lead to less homogeneous and porous films [37]. In order to deposit homogeneous CS-metal coatings without the formation of additional inorganic phases, as well as to take advantage of the properties of CS-metal ion complexes, a two-step EPD coating procedure has been recently demonstrated for the CS-Cu (II) complex [38]. The success of the applied method lies in the fact that soluble polymer-metal ion complexes behave as polyelectrolytes in aqueous solutions, so continuous and crack-free coatings could be prepared. Nevertheless, the cited study showed that high concentrations of copper ions in CS-Cu (II) complex coatings gave rise to cytotoxic effects in human osteoblast-like cells. Therefore,

the choice of a therapeutic metal ion in the chitosan complex must be optimized to achieve the desired biocompatibility of the final coating. In this context, the Ga ion is a potential candidate due to its comparable low toxicity and good antibacterial efficacy.

The purpose of this work was, thus, to prepare Ga (III)-CS complex coatings on stainless steel substrates by EPD following a two-step coating procedure. By taking the benefits of CS chelation properties, robust homogenous coatings with uniform Ga(III) distribution were synthesized, which resulted in improved physical and biological performance of the coatings. To the best of the authors' knowledge, this type of Ga(III)-CS coatings has not been previously investigated for potential biomedical applications.

## 2. Experimental

### 2.1. Materials

Chitosan (CS) of medium molecular weight with a ~75%–85% degree of deacetylation was used from Sigma-Aldrich, Taufkirchen, Germany, which had already been used to obtain coatings by EPD [38–40]. Gallium (III) nitrate hydrate and acetic acid (99%) were purchased from Sigma-Aldrich, Germany. Ethanol (99%) and PBS tablets were obtained from VWR, Darmstadt, Germany.

### 2.2. Preparation of EPD Suspensions

Initially, the preparation of the Ga (III)-CS complex was performed by modifying a previously described protocol [12]. Briefly, chitosan (2% w/v) was dissolved in a 2% *v/v* acetic acid solution. Subsequently, a gallium source ($Ga(NO_3)_3 \cdot nH_2O$) was added at different ratios. The precipitation of Ga (III)-CS was achieved by adding the complex solution dropwise in 99% ethanol. After 2 h, the final solid products were washed several times until complete pH neutralization and the product was then dried at 60 °C overnight.

To prepare the EPD suspensions, 1 g/l Ga (III)-CS complex was dissolved in 20 vol. % deionized water and 1 vol. % acetic acid by magnetic stirring. After dissolution, the volume of the solution was filled to 100% with ethanol under stirring. The used concentration of Ga (III)-CS in the solvent was optimized using the Taguchi experimental design approach. At the same time, the diluted acetic acid-ethanol solvent was chosen, in agreement with already reported results [1]. The pH of the suspension before EPD was adjusted to be 4.5. Labelling of the coatings was performed according to the ratio of Ga (III) to CS free amino groups ($Ga^{3+}:NH_2$), namely pure CS coatings and those with Ga (III) concentrations of 1:32, 1:16, 1:8, and 1:4 were investigated.

### 2.3. Deposition Procedure

Electrophoretic deposition was performed at a fixed voltage level of 15 V, applied by a Thurlby Thandar Instruments (TTi) EX752M power supply (Huntingdon, UK). The deposition time was set to 10 min, while the distance between the parallel electrodes was kept constant at 1 cm (the deposition parameters were determined by a statistical optimization method). AISI 316L stainless steel foil of 0.2 mm thickness was cut into plates with a width of 1.5 cm and a length of 3 cm to prepare the deposition electrode and counter-electrode. Substrates were then cleaned with an ethanol/acetone (1:1 vol. %) mixture under ultrasonication, washed with deionized water, and air-dried prior to deposition. The deposition area was fixed at 2.25 $cm^2$ by submerging 1.5 cm of the substrate length in the solution. After coating, the samples were left to dry at room temperature before further characterization. In vitro cell culture tests were performed on coated circular AISI 316L substrates with a diameter of 1.5 cm and a thickness of 0.1 mm. Such circular samples were produced using the same parameters used for rectangular ones.

### 2.4. Suspension Characterization

The zeta potential of all Ga (III)-CS suspensions was measured by Laser Doppler Velocimetry (LDV) technique, using a Zetasizer Nano (Malvern Instruments, Malvern, UK). Moreover, the influence of Ga (III) concentration in suspension on the variation of current density versus deposition time was recorded and monitored by a TTi 1906 Computing Multimeter.

### 2.5. SEM and EDX Spectroscopy

Surface and cross-section morphologies of the prepared complex coatings were analyzed by scanning electron microscopy (SEM, Auriga CrossBeam, Carl Zeiss Microscopy GmbH, Jena, Germany). Specimen preparation before examination included sputter-coating with a Q150T S, equipped with a gold target (Quorum Technologies Ltd., Lewes, UK). Coating thickness was calculated using ImageJ image processing software (Java, National Institutes of Health, Bethesda, MD, USA), as an average of 10 measurements from different captured areas. Furthermore, an energy dispersive X-ray (EDX) spectrometry detector combined with SEM was employed to detect the elemental distribution in selected areas of the samples. EDX data were obtained at an electron accelerating voltage of 20 kV.

### 2.6. FTIR Analysis

The chemical structure of all coatings was analyzed qualitatively by Fourier transform infrared spectroscopy (FTIR), using a Shimadzu IRAffinity-1S spectrometer (Shimadzu Corp., Tokyo, Japan) equipped with LabSolution IR software (version 2.12). The data were collected in absorbance mode in the wavenumber range of 400–4000 $cm^{-1}$ at a resolution of 4 $cm^{-1}$.

### 2.7. Contact Angle Measurements

Contact angle measurements were conducted with a DSA30 instrument (Kruess GmbH, Hamburg, Germany) to evaluate the wettability of the coatings. For this purpose, deionized water droplets were placed on the surface of the substrates and the evolution of the droplet shape was recorded with a video camera over a time period of 5 min. The procedure was repeated at least three times in different positions on the sample.

### 2.8. Swelling Study

The swelling capability of Ga (III)-CS coatings was determined by immersing the coated stainless steel substrates in 10 mL phosphate buffered saline (PBS) solution for 1, 3, 7, 10, 15, 20, 25, 30, 40, 50, 60, 90, and 120 min. Three samples of each system were analyzed. The swelling ratio was determined by measuring the weights of the dry sample ($W_d$) and the wet sample immediately after removal from the solution ($W_w$), according to the following formula:

$$Swelling \ [\%] = \frac{W_w - W_d}{W_d} \times 100\% \tag{1}$$

### 2.9. In Vitro Degradation Study

A degradation study of the coatings was performed in PBS (pH = 7.4), containing 1.5 µg/mL lysozyme at 37 °C in a shaker incubator (79 rpm). The lysozyme concentration was selected, taking into account its concentration in human serum [41]. After incubation at predetermined time intervals of 1 h, 3 h, 7 h, 24 h, 3 days, and 7 days, the samples were removed from the solution and dried at 37 °C overnight. The experiment was conducted in triplicate. The degradation (%) was calculated from the weights of dry coated samples before degradation ($W_1$) and dry coated samples after degradation ($W_2$), respectively, as:

$$Degradation \ [\%] = \frac{W_1 - W_3}{W_1} \times 100\% \tag{2}$$

## 2.10. Mechanical Characterization

The variations in the hardness (H) of the coatings as a function of their composition were assessed by nanoindentation tests, using a Zwick/Roell ZHN Universal Nanomechanical Testing System (Ulm, Germany). A pyramidal diamond indenter was employed for the indentation experiments. All samples were mounted on flat aluminum stubs with super glue. The load was held at a maximum value of 5 mN for 60 s. Twenty indentations were performed on each sample, as each indent was exactly 100 μm away from the other to avoid interactions between the created plastic strain fields. Hardness values were obtained using the Zwick/Roell ZHN software (Ulm, Germany).

The scratch resistance of Ga (III)-CS coatings was measured using a CSM Instruments scratch tester (recently Anton Paar, Graz, Austria). A Rockwell diamond tip with a tip radius of 200 μm was used to produce controlled scratches ($n = 3$) with a length of 5 mm under linear progressive load ranging from 1 to 10 N, with a loading rate of 3.6 N/min and speed of 2 mm/min. The critical load at which the coatings started to fail was detected with optical photographs. SEM was also performed for high-resolution imaging of the scratched surfaces.

## 2.11. Antibacterial Assay

The effect of Ga (III) incorporation in the CS coatings on the viable counts of Gram-negative *Escherichia coli* was investigated using the colony forming unit test, considering pure CS coatings as a control. The experiment was performed with little modification from a previously reported method by Yuan et al. [42]. Prior to the experiment, all samples were sterilized using UV treatment for 1 h. Isolated colonies of the test strains were cultured in a nutrient broth (lysogeny broth (LB) medium, Carl Roth GmbH, Karlsruhe, Germany) overnight at 100 rpm and 37 °C in order to obtain a fresh bacteria suspension suitable for inoculation. On the second day, the fresh bacteria suspension was diluted to an optical density (600 nm, Thermo Scientific™ GENESYS 30™, Schwerte, Germany) of 0.015. Thirty microliters of bacterial suspension of ~$1 \times 10^7$ colony forming units (CFU)/mL bacteria was dropped onto the sample surface and cultured for 24 h at 37 °C. The samples were then transferred into sterilized centrifuge tubes containing 3 mL of LB broth, then the tubes were agitated intensely for 30 s to detach the bacteria from the sample surfaces. The detached bacteria suspension was diluted $10^3$ times with LB broth. A volume of 30 microliters of diluted bacterial suspension was evenly spread onto a fresh agar (Luria-Bertani (LB) agar, Carl Roth GmbH, Karlsruhe, Germany) plate. After 24 h of culture at 37 °C, the live bacteria cells were counted according to the National Standard of China GB/T 4789.2-2010 protocol [43]. The antibacterial percentage (P) was calculated from the average number of bacteria colonies on the CS coating (*A*) and on the Ga (III)-CS complex coating (*B*), using the following formula:

$$P\ [\%] = \frac{A - B}{A} \times 100\% \tag{3}$$

## 2.12. In Vitro Cellular Test

### 2.12.1. Cell Culture and Seeding

MG-63 cells, a human osteosarcoma cell line, were used to assess in vitro cytocompatibility of the coatings. This cell line is commonly applied to characterize biomaterials intended to be in contact with bone, because MG-63 cells express many of the characteristics of normal osteoblasts [44]. Tissue culture plate was used as control. Prior to the experiments, all coatings were placed in a 24-well plate and sterilized under UV light for 1 h. Cells were cultured in cell culture polystyrene flasks using Dulbecco's modified Eagle's medium (DMEM, Gibco, Schwerte, Germany), supplemented with 10 vol. % fetal bovine serum (FBS, Sigma-Aldrich) and 1 vol.% penicillin/streptomycin (Pen-Strep; Sigma-Aldrich). Once the confluency reached 80%, the monolayer of cells was detached from the flask wall with trypsin–EDTA solution (Life Technologies, Darmstadt, Germany) in PBS. Upon cell detachment, trypsination was inactivated by adding fresh DMEM and the cell suspension was counted

in a hemocytometer by trypan blue exclusion method (Sigma-Aldrich, Taufkirchen, Germany). A cell suspension of $10^5$ cells per mL was prepared and 1 mL of it was used to directly cover the sterilized samples, which were also preconditioned in DMEM for 1 h to achieve initial protein adhesion on their surfaces. Cells were then allowed to grow on the coatings and control samples for 24 h at 5% $CO_2$ and 37 °C.

### 2.12.2. Cell Viability

The viability of MG-63 cells was quantified by the water-soluble tetrazolium salt (WST-8) test. After an incubation period of 24 h, the old culture medium was discarded and the samples were washed with PBS. Then, 1 mL fresh DMEM containing 1 vol. % WST-8 reagent was added in each well. After 3 h of incubation, 2 aliquots of 100 μL from each sample were transferred in a 96-well plate and the absorbance at 450 nm was measured with a micro plate reader. Two independent experiments were conducted with six samples for each formulation. The cell viability (%) was computed from the optical density ($OD$) of each specimen ($OD_{sample}$), the WST reactant ($OD_{blank}$), and the respective positive control ($OD_{reference}$):

$$Cell\ viability\ [\%] = \frac{OD_{sample} - OD_{blank}}{OD_{reference} - OD_{blank}} \times 100\% \tag{4}$$

### 2.12.3. Cell Morphology and Attachment

In order to qualitatively evaluate the cell morphology and viability, live staining with DAPI (4′,6-diamidino-2-phenylindole) and Calcein AM (Life Technologies, Darmstadt, Germany) was applied, following the protocols provided by the supplier. The images were taken with a fluorescence microscope (FM) (Axio Scope A1, Carl Zeiss Microimaging GmbH, Jena, Germany).

### *2.13. Statistical Analysis*

The experimental results are represented as the mean value ± standard deviation (SD) for each group of samples. Statistical analysis was performed using one-way ANOVA test with Bonferroni's post-hoc test, with a probability of $p < 0.05$ considered as being statistically significant.

## 3. Results and Discussion

### *3.1. Influence of Ga (III) Concentration on Suspension Properties*

As shown in Table 1, all suspensions exhibited similar zeta potential values irrespective of Ga (III) concentration, which is explained by the identical suspension parameters (pH, suspension medium, etc.). In addition, the positive charging predicted the (cathodic) deposition of the Ga (III)-CS complex on the negatively charged electrode, which is typical of electrodeposition of CS [31,45].

**Table 1.** Zeta potential of chitosan (CS) and gallium (Ga (III))-CS complex suspensions used for electrophoretic deposition (EPD).

|  | CS | 1:32 | 1:16 | 1:8 | 1:4 |
|---|---|---|---|---|---|
| **Zeta-Potential [mV]** | 32 ± 6 | 29 ± 7 | 27 ± 7 | 31 ± 7 | 28 ± 6 |

Pure CS and Ga (III)-CS suspensions with concentrations of 1:32, 1:16, and 1:8 exhibited similar magnitudes of the current densities during EPD, ranging from 0.0019 to 0.0018 A/cm². However, at the highest Ga (III) concentration (1:4), the absolute values of current density drastically increased to 0.0028 A/cm². This phenomenon could be associated with the fact that the main current carriers in the suspensions are the free ions, so that the increased amount of Ga (III) ions leads to a higher conductive flow [46]. Further analysis (see below) confirmed that the higher current density of 1:4 Ga (III)-CS suspensions was a decisive factor affecting the morphological characteristics of the coatings.

It should be emphasized that the pH of the suspensions before EPD was intended to be 4.5, in order to facilitate the solubility of CS and to increase its metal-binding ability. On the one hand, in highly acidic solutions the adsorption capacity of CS for cations is lowered because $H^+$ ions compete with metals to bind to the polymer active sites [47,48]. Additionally, the protonation of amino groups in acidic environments induces the electrostatic repulsion of metal cations, which also limits the chelation performance [47]. On the other hand, at neutral to basic pH, the amine groups in the glucosamine monomer unit of the polysaccharide structure deprotonate. As a result, the lone pair of electrons in the CS functional group presents a preferable site for chelation with electropositive metal ions such as gallium. Still, the increasing deprotonation of CS is also related to lower electrostatic repulsion between monomeric units, which could lead to aggregation and instability of the suspension during deposition. Thus, it is hypothesized that after dissolution of the Ga (III)-CS complex in the prepared slightly acidic medium, only a limited number of amine groups becomes protonated [49] and the Ga (III) ion still remains bound within the covalent coordination complex with CS. Under the applied voltage, the positively charged CS, together with the metal cation gallium, is attracted toward the negative electrode. Due to the alkaline environment around the cathode created by the accumulation of hydroxyl ions, Ga (III) and CS deposit on the cathode in the form of a polymer-metal ion complex. Since Ga (III) exists preliminary in bound form within the CS matrix, the reduction of free metal ions by the available electrons on the electrode can be avoided. Thus, the formation of new crystalline species or metallic Ga particles is hindered, which is highly important to control the release of Ga ions upon degradation of the chitosan matrix in certain applications.

## 3.2. Morphological and Compositional Characterization

The microstructure of CS and Ga (III)-CS coatings at low and high magnifications is presented in Figure 1. Due to the similar morphological features of CS coatings and 1:32, 1:16, and 1:8 Ga (III)-CS coatings, only data for CS samples are shown, as an example. SEM images of all deposits under low magnifications indicated that the coatings had smooth, homogeneous, crack-free surfaces. No crystalline structures were detected, which confirmed the formation of a coordinated complex between CS and Ga (III). Similar observations have been reported for electrophoretically deposited Cu (II)-CS coatings with the same concentration of copper ions [38]. Despite the comparable morphology of all Ga (III)-CS coatings at the microscale, high magnification images revealed that addition of Ga (III) in the highest concentration of 1:4 led to the formation of multiple (nano)cracks on the sample surface. The explanation could be found in the complexation of $Ga^{3+}$ ions with CS through $-NH_2$ and $-OH$ binding sites, which reduces the number of intermolecular and intramolecular hydrogen bonds in the polymer [50]. The excess number of Ga (III) ions involved in the process of complex formation can lead to the disruption of the regular structure of CS chains, which can induce the formation of cracks on the coating surface [50]. In addition, the deposition rate of the polymer molecules affects their packing in the coating. As already stated, 1:4 Ga (III)-CS suspensions showed higher current density during EPD in comparison to the other formulations. Due to the high electrophoretic mobility, molecules move rapidly, meaning that they do not have sufficient time to settle down in their suitable positions to form a close-packed structure, which is similar to the behavior observed in EPD from particle suspensions [51].

The current density affected not only the surface morphology, but also the coating thickness (Figure 1C,F). Pure CS coatings and those with Ga concentrations of 1:32, 1:16, and 1:8 exhibited thicknesses in the range of 40.6 ± 0.4 μm. By contrast, the significantly higher current density for 1:4 Ga (III)-CS suspensions led to excessive accumulation of $OH^-$ ions in the cathode, which neutralized more CS molecules [52]. Accordingly, a proportional increment of the coatings thickness to 100 ± 0.8 μm was observed for 1:4 Ga (III)-CS coatings.

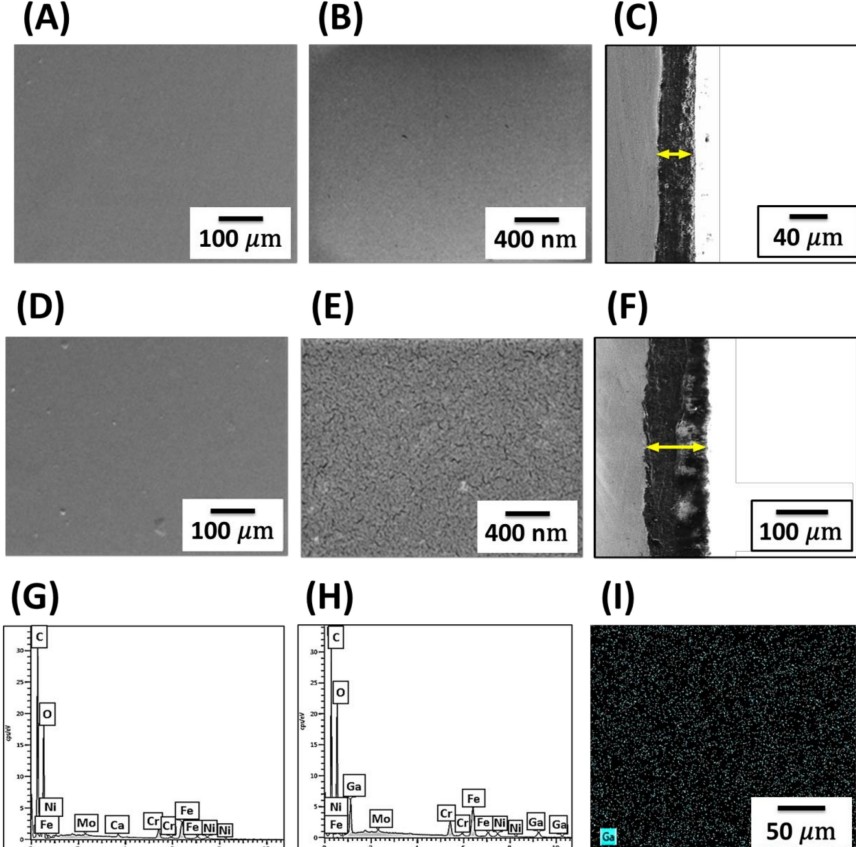

**Figure 1.** SEM micrographs of the surface of the (**A,B**) CS coating, (**D,E**) 1:4 Ga (III)-CS complex coating, and cross-section morphologies of the (**C**) CS coating and (**F**) 1:4 Ga (III)-CS complex coating. EDX spectra of the (**G**) CS coating, (**H**) 1:4 Ga (III)-CS complex coating, and (**I**) elemental EDX mapping of Ga for the 1:4 Ga (III)-CS complex coating.

EDX spectra of all investigated coatings (Figure 1G,H) showed the presence of carbon and oxygen, which are constituting elements of CS, while the metallic ions (Fe, Cr, Ni, Mo, Ca, and Mn) can be attributed to the stainless steel substrate [12,53,54]. Additionally, the presence of metal ions in all complex coatings was demonstrated by the gallium K and L peaks at 10.2, 9.2, and 1.0 keV, whose relative intensity increased with the increasing quantity of doped ions. Ga elemental mappings across the surfaces indicated uniform distribution of the metal ions in the polysaccharide matrix for all Ga (III)-CS coating types. Since similar results were obtained with all Ga-containing sample types, only data for the 1:4 Ga (III)-CS coating are shown (Figure 1I). No contamination was detected from laboratory tools or residual salts. The amount of Ga (III) loaded in the coatings was semi-quantitatively evaluated by normalizing the area of the gallium L peak with respect to the carbon peak from four different EDX spectra of each formulation (Table 2). The same estimation of the Ga (III) quantity was performed for the CS-Ga (III) complex before dissolution in the EPD bath. It is clear that with increasing concentration of added Ga (III) in the form of the Ga (III)-CS complex, the resulting $Ga_{kL}/C_{k\alpha}$ ratio in the coatings also increased, confirming the chelation of Ga (III) ions [12]. The amount of Ga (III) in the coatings was lower than the Ga (III) quantity in the as-prepared material of identical concentration, because after dissolution of the complex in the EPD solvent a reduced number of Ga (III) ions actually deposited on the cathode. Furthermore, the calculated $Ga_{kL}/C_{k\alpha}$ ratio in the as-prepared Ga (III)-CS relative to the $Ga_{kL}/C_{k\alpha}$ ratio in the coatings remained nearly constant, which confirmed the reproducibility of EPD in developing these novel types of CS-based coatings.

**Table 2.** Intensity ratios for the gallium L peak and carbon peak from EDX spectra of the as-prepared Ga (III)-CS complex and Ga (III)-CS coatings.

| Sample | 1:32 | 1:16 | 1:8 | 1:4 |
|---|---|---|---|---|
| $Ga_L/C_{k\alpha}$ ratio of Ga (III)-CS (%) | 3.1 ± 0.2 | 7.1 ± 0.3 | 10.5 ± 0.4 | 32.1 ± 0.9 |
| $Ga_L/C_{k\alpha}$ ratio of Ga (III)-CS coating (%) | 2.4 ± 0.1 | 5.1 ± 0.1 | 8.0 ± 2.9 | 24.2 ± 0.1 |

Figure 2 shows the comparison between the FTIR spectra of bare CS and Ga (III)-CS coatings. No measurable differences are observed in the spectra of samples with different amounts of Ga (III); for clarity reasons only data for the 1:4 Ga (III)-CS samples are shown. Pure CS coatings exhibit a broad band in the range of 3400–3200 cm$^{-1}$, which corresponds to the overlapping of the stretching vibration of –NH$_2$ and asymmetrical stretching vibration of –OH [55]. The peak at 2875 cm$^{-1}$ is assigned to the characteristic C–H asymmetric stretch vibrations of –CH$_2$ [15]. CS also showed a peak at 1650 cm$^{-1}$, which is attributed to the stretching of the C=O group of the amide I band, while the amide II band at 1550 cm$^{-1}$ accounts for –NH bending vibrations [12,15]. The peak at around 1400 cm$^{-1}$ is characteristic of either the deformation of C-H or for the stretching of C–N [12], while the band at 1375 cm$^{-1}$ can be assigned to the CH$_3$ symmetrical deformation mode [56]. The amide III band at 1320 cm$^{-1}$ represents C–N stretching vibration [57]. The absorption bands in the range of 1150–1000 cm$^{-1}$ are associated with C–O and C–O–C bands of the polysaccharide structure of CS, and the CH deformation of the β-glycosidic bond is centered at 895 cm$^{-1}$ [57].

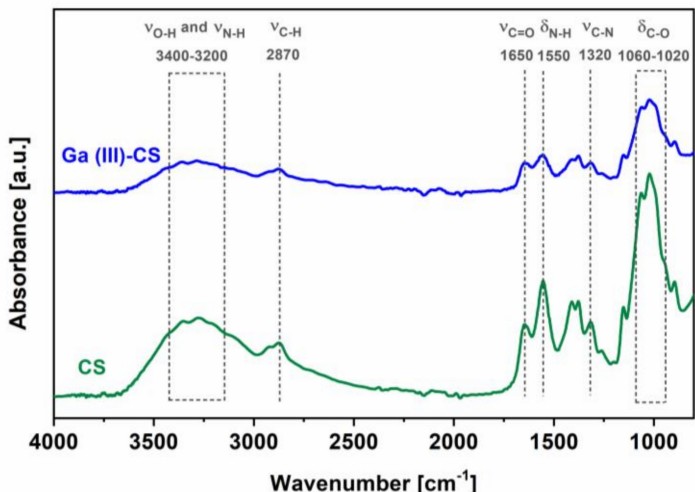

**Figure 2.** FTIR spectra of CS and Ga (III)-CS complex coatings (1:4). Spectra of all Ga (III)-containing sample types are comparable.

CS-metal ion complex coatings showed basically the same spectra as their precursor polymer, with several key bands modified. For all Ga (III)-CS complex derivatives, a decrease in the relative absorbance of the band at 3300 cm$^{-1}$ associated with $v_{O-H}$ and $v_{N-H}$ was observed. This can be explained by the fact that the interaction between CS and the Ga (III) metal ion causes a decrease in the electron density of –NH$_2$ and –OH and in the bond force constant of CS chemical bonds, suggesting that both amine and hydroxyl groups take part in chelation [12,58]. The complex formation was also indicated by the decrease in absorbance for amide and amine bands at 1650 and 1550 cm$^{-1}$ [12]. Changes related to C–CH$_3$ deformations in amide groups were observed at the frequency of 1320 cm$^{-1}$ [59]. The peaks for C–O and C–O–C bonds at 1062and 1020 cm$^{-1}$, respectively, changed as a consequence of the stretching of glucosamine monomers of CS chains by the cross-coordination with the Ga (III) ion [12]. The modification of the shape and intensity of the C–H stretching bands at 1409 and 2873 cm$^{-1}$ likely

resulted from a modification in the $CH_2OH$ environment due to the presence of Ga (III) intercalated in the structure of CS [60].

### 3.3. Wettability

The surface of the biomaterial is the first component of the implant that comes into contact with the biological environment. Shortly after implantation, biomaterials are covered by a layer of plasma proteins, such as albumin, fibrinogen, IgG, and fibronectin [61]. Cells generally adhere to the surface of the material via interactions with this layer of adsorbed proteins [62]. Thus, wettability, which describes the ease of fluid spreading across a solid surface, is an important factor that determines protein adsorption and subsequent cellular response [63]. Wettability is evaluated by measuring the contact angle at the liquid–solid surface. Figure 3A shows the contact angle values of the coatings with different Ga (III) ion concentrations as a function of droplet deposition time. CS and Ga (III)-CS complex coatings at 1:32, 1:16, and 1:8 concentrations exhibited initial contact angles in the range of 96°–100°. The addition of a higher amount of Ga (III) ions in the 1:4 sample reduced the starting contact angle significantly to 89° ± 2°. This measurement is in line with a qualitative assessment of the droplet profiles (Figure 3B). The explanation can be attributed to the presence of multiple nanocracks on the surface of these coatings (see Figure 1E), which induce a rapid infiltration of water into the inner part of the coating [12]. With a prolonged time for the deposition of droplets on the surface, the contact angles for all systems reduced because of the liquid dispersion over the area. Although high wettability usually facilitates faster initial cell adhesion to surfaces [12], hydrophobic substrates promote fibrinogen adsorption, albumin adsorption, and platelet adhesion [64]. Therefore, the CS-Ga (III) coatings developed here are expected to allow specific protein attachment, which will be beneficial for long-term cell–surface interactions [12].

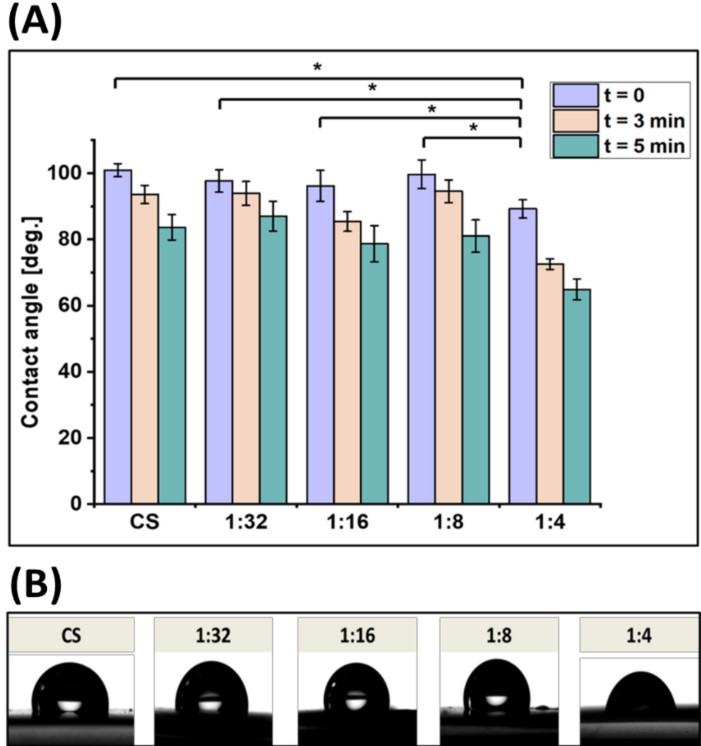

**Figure 3.** (**A**) Average contact angle of water droplets on CS and Ga (III)-CS complex coatings according to Ga (III) content, measured at three time points (i.e., immediately after deposition, after 3 min, and after 5 min). Asterisks (*) denote significant differences ($p < 0.05$). (**B**) Profiles of water droplets on CS and Ga (III)-CS complex coatings with increasing quantities of Ga (III) (from left to right) immediately after deposition.

### 3.4. Physical and Mechanical Properties

The swelling behavior of CS and Ga (III)-CS complex coatings was determined in PBS solution at progressive intervals up to 120 min. The swelling curves are displayed in Figure 4A. It was observed that all coatings experienced equilibrium in terms of absorbing PBS solution at around 40 min after immersion. After this time point, the swelling ratio remained approximately constant for the whole course of the study. The maximal swelling ratio for pure CS coatings was measured to be 9.8% ± 0.6% after 60 min. After inclusion of the Ga (III) ion, the maximum swelling capacity of the complex coatings generally decreased, as this trend was highlighted as having a higher Ga (III) ion concentration. The incorporation of an increasing amount of metal ions in the CS matrix may have hindered polymer chain mobility restricting water penetration into the network, which was mirrored in the reduced swelling effect [65]. This detected swelling variability with Ga (III) concentration could play a crucial role in the design of implant materials, because it can regulate the transfer of cell nutrients and the absorption of physiological fluids in the human body. However, a further consideration of this topic was beyond the scope of the present study.

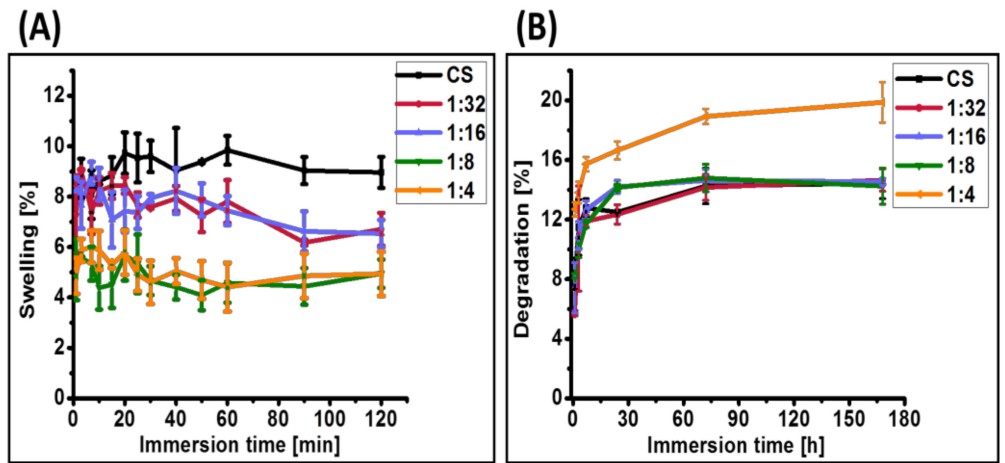

**Figure 4.** (**A**) Swelling study in PBS and (**B**) degradation study in PBS–lysozyme of CS and Ga (III)-CS complex coatings with different Ga (III) concentrations.

Figure 4B shows the weight loss of the coatings as a function of incubation time in the lysozyme–PBS solution at various time intervals. No significant variations in degradation were observed for pure CS coatings or with Ga (III) concentrations of 1:32, 1:16, and 1:8, as their degradation rate became stable after 24 h. Loading the CS coatings with the highest amount of Ga (III) (1:4 concentration) led to an increased coating degradation. It can be anticipated that due to the higher nanoporosity of these coatings, a greater contact surface area with the biological medium results. Consequently, lysozyme can easily infiltrate into the Ga (III)-CS matrix and break the glycoside bonds, which are a major target for hydrolytic scission by lysozyme. Another explanation is that the coordination bond between CS and metal ions causes some weak points in the CS chain, which can be broken easier than those of CS itself [66].

The degradation pattern of high initial mass loss followed by a reduced degradation rate, which was observed for all types of coatings, can be attributed to the specific action mechanism of the enzyme. Lysozyme is able to hydrolyze the glycosidic linkages between N-acetyl-glucosamine units in the CS structure [67]. The enzyme contains an hexasaccharide sequence with 3–4 or more acetylated units, which is mainly responsible for the initial degradation rate of N-acetylated CS [68]. However, with progressing degradation, the appropriate hexasaccharide sequences in lysozyme are lost, which leads to a slower rate of weight loss [68].

Results from nanoindentation experiments of CS and Ga (III)-CS coatings are presented in Figure 5. It is apparent that the addition of Ga (III) metal ions at 1:32, 1:16, and 1:8 concentrations gradually

increased the hardness of the polymer coating. The reason is likely related to the effect of chelation, which restricts the mobility and deformation of the CS matrix by introducing a local resistance to indentation [69]. The same behavior has been reported in the previous study on electrophoretic Cu (II)-CS coatings [38]. Interestingly, it can be seen that in the case of Cu (II)-CS coatings, there was no drastic change in hardness with the change in copper concentration, as compared to the Ga (III)-CS coatings presented here. This result is likely due to the difference in bonding behavior of divalent and trivalent ions. As copper is divalent, it forms tetra-coordinative complexes, while Ga(III) forms a hexa-coordinative complex [13,70]. In this way, Ga (III)-CS involves more amine and hydroxyl bonds in the complex inducing a higher rigidity of the CS matrix. Nevertheless, the dose-dependent increase in hardness with higher Ga (III) concentration was not valid for coatings chelated with the largest amount of metal ions, which might be explained by the presence of cracks on the surface of the samples (Figure 1E) [50]. Moreover, Qu et al. reported that the mechanical properties of CS increase up to a certain limit by complexation [50]. However, at high concentrations, mechanical properties start to decrease due to crack formation caused by the involvement of too many metal ions in the complex [50]. According to the literature, a broad range of nanohardness values are reported for chitosan, as nanoindentation values depend strongly upon different factors and experimental conditions, including indenter size, shape, and indentation depth [69,71,72].

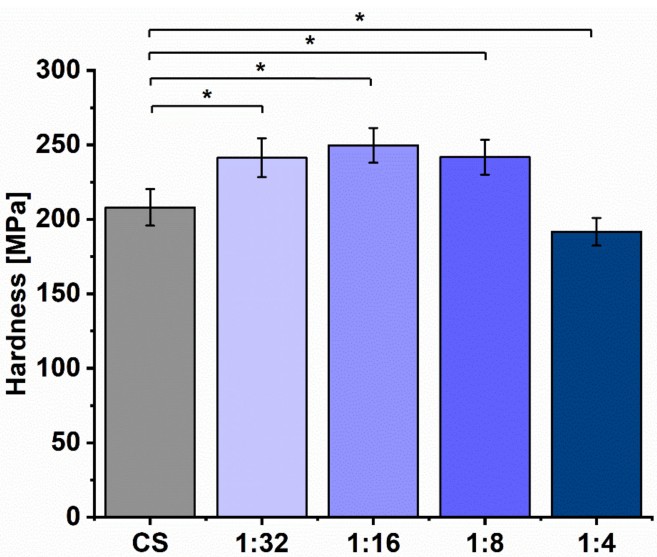

**Figure 5.** Comparison of the hardness of CS and Ga (III)-CS complex coatings with different Ga (III) concentrations. Asterisks (*) denote significant differences ($p < 0.05$).

A scratch test was conducted to determine the critical loads of the coatings. The critical load can be defined as the minimum load at which the coating ruptures and exposes the substrate [73]. This value is used to compare the cohesive or adhesive properties of coatings.

Table 3 shows the critical loads of CS and Ga (III)-CS coatings. It was noted that all Ga-containing coatings showed a higher critical load compared to the CS coating. However, at lower concentrations of Ga (III) (1:32 and 1:16) the critical load increased significantly, while at higher concentrations a decrease in the critical load can be observed. This behavior might be due to the fact that lower concentrations of the Ga (III) synergistically increase the adhesive properties of the polymer coating to the substrate. However, as already described, high concentrations of Ga (III) impair the regular structure of the CS matrix and induce cracks on the coating surface (Figure 1E). Stress concentrates on the crack tips, leading to lower crack resistance ability of the coatings under load, expressed macroscopically by the lower measured critical loads.

**Table 3.** Critical load values obtained by scratch test with respect to the standard deviation of CS and Ga (III)-CS coatings.

| Sample | CS | 1:32 | 1:16 | 1:8 | 1:4 |
|---|---|---|---|---|---|
| Critical Load (N) | 2.2 ± 0.1 | 3.1 ± 0.1 | 3.1 ± 0.2 | 2.7 ± 0.1 | 2.6 ± 0.1 |

Figure 6 shows micrographs of scratches on the coatings at low (1 N) and high (10 N) loads. No significant difference in these micrographs can be observed. At lower loads, CS and complex coatings with low concentrations of Ga (III) showed plastic deformation before peeling off from the substrate. However, 1:8 and 1:4 Ga (III)-CS samples showed delamination of the coatings without plastic deformation.

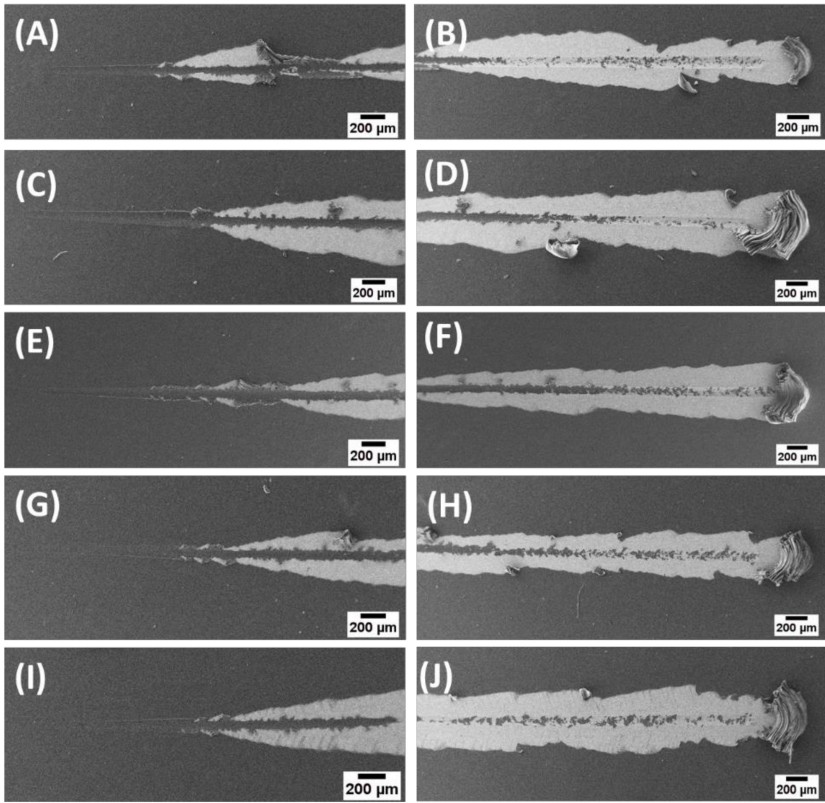

**Figure 6.** SEM micrographs of scratches at low (1 N) and high (10 N) loads on (**A,B**) CS, and on Ga (III)-CS coatings at (**C,D**) 1:32, (**E,F**) 1:16, (**G,H**) 1:8, and (**I,J**) 1:4 concentrations.

*3.5. Antibacterial Assay*

The antibacterial activity of the Ga (III)-CS complex coatings against *E. coli* after 24 h of incubation is presented in Figure 7. CS coatings were used as control, equating their antibacterial activity to 0%. It could be concluded that the average antibacterial percentage of all coating types against the Gram-negative strain exceeded 90% compared to the control of pure CS. The reason likely lies in the complexation between CS and metal ions. It is known that chelation increases the positive charge density of CS, leading to enhanced adsorption of polycations onto the negatively charged cell surface, disruption of the bacterial cell envelope, and bacteria death [14]. On the other hand, Ga (III) also exerts a deleterious effect against *E. coli* due to its ability to interact with iron uptake systems. Even though the chemical similarities of $Ga^{3+}$ with $Fe^{3+}$ allow its transportation inside the bacteria cell, Ga cannot participate in iron-catalyzed reduction–oxidation reactions. Consequently, microbial death commences due to the damage of DNA molecules, production of reactive oxygen species (ROS), or disruption of other essential metabolic processes [18,74].

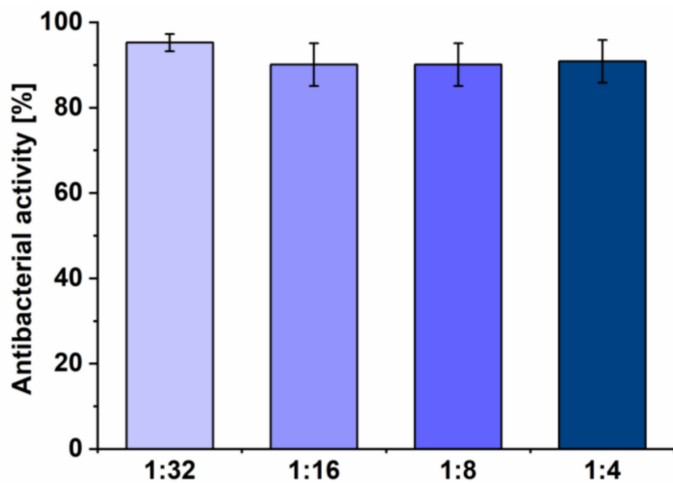

**Figure 7.** Antibacterial activity of Ga (III)-CS coatings with different Ga (III) concentrations relative to pure CS coatings against *Escherichia coli* after 24 h of incubation. Statistical analysis was performed with respect to bare CS samples.

### 3.6. Biological Characterization

The cellular activity of MG-63 cells cultured on the coatings for an initial period of 24 h was examined by measuring cell viability and cell attachment. The results show statistically significant reductions of 69% ± 12% in cell mitochondrial activity in the samples with a bare CS layer relative to the control (Figure 8B). Since CS has been reported to be non-toxic to MG-63 cells [75], the reason for this decreased viability could be related to the relatively low number of attached cells. It is common knowledge that cell adhesion first involves non-specific forces, such as electrostatic or van der Waals forces, followed by specific (i.e., receptor-mediated) interactions of the cell with the substrate [76]. Non-specific electrostatic interactions between protonated amine groups from the glucosamine unit in CS and negatively charged carboxylate and sulfate groups in cell surface proteoglycans are responsible for cell adhesion to CS [77]. However, CS lacks specific binding domains, which are mediated by integrin receptors, or cell recognition sequences (such as RGD) that promote cell adhesion [78,79]. As a consequence, a lower number of cells could attach on the surfaces of CS-based coatings in comparison to the control. In addition, the hydrophobic nature of CS, which is related to low surface energy, may also cause lower cell affinity [80]. However, the similar cell viability values of CS coatings to 1:32, 1:16, and 1:8 Ga (III)-CS coatings reflect that the presence of Ga (III) did not increase the cytotoxicity level of the material. Cell viability measurements coincide with the fluorescence microscopy observations (Figure 8A) in that all CS-based coatings showed slightly decreased amounts of cells on their surfaces compared to the control. There is no appreciable difference in MG-63 cell morphology in the samples with varying Ga (III) ion concentration, confirming the non-toxic character of the complex coatings.

The obtained Ga (III)-CS coatings showed superior cell viability in comparison to the previously prepared Cu (II)-CS complex coatings [38]. In addition, osteoblast cells exhibited better attachment and spreading on the Ga (III)-CS coatings than on the Cu (II)-CS ones reported elsewhere [38].

A similar finding was also reported in another study, where gallium revealed the lowest cytotoxicity to human fibroblasts among various metal ions released from dental alloys, including copper, zinc, silver, indium, and mercury [81]. Indeed, there is a threshold of Cu (II) ion concentration, above which the Cu (II)-CS complex damages eukaryotic cells [12]. In contrast, our results on Ga containing CS demonstrate a highly beneficial system for the development of antimicrobial materials with suitable biocompatibility, because even at the highest tested Ga content the Ga (III)-CS coatings showed a lack of toxicity whilst maintaining outstanding antibacterial behavior.

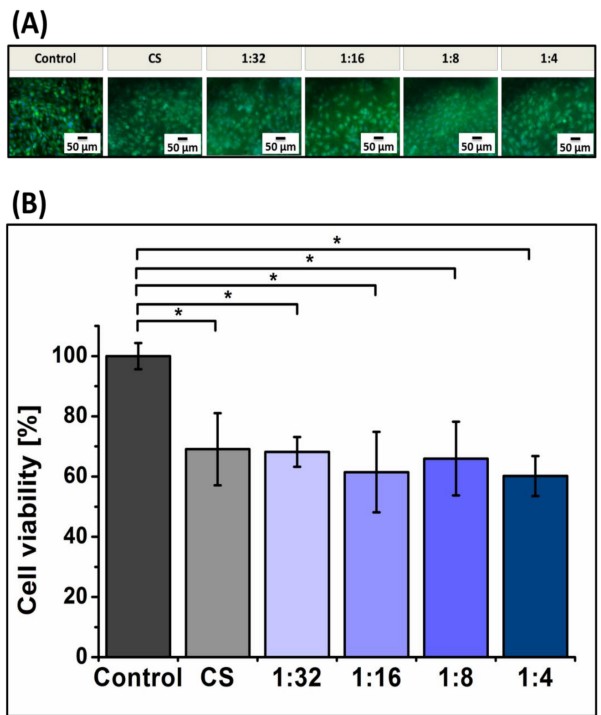

**Figure 8.** (**A**) Fluorescence microscope images, showing the results of Calcein-DAPI staining of MG-63 cells after 24 h of culture on CS and Ga (III)-CS complex coatings with different Ga (III) concentrations. (**B**) Cell viability of osteoblast-like cells (MG-63) cultured on CS and Ga (III)-CS complex coatings with different Ga (III) concentrations after 24 h of incubation. Tissue culture plate was used as a control. Asterisks (*) denote significant differences ($p < 0.05$).

## 4. Conclusions

The results of this investigation pave the way for the synthesis of homogeneous Ga (III)-CS complex coatings (without precipitation of metallic gallium) on stainless steel substrates by EPD. One relevant observation is that the adjustment of the Ga (III) ion amount embedded in the CS matrix can be used to tune the morphological, physical, and mechanical properties of the coatings. This approach could contribute to the generation of "tailor-made" CS-based coatings for implants. Moreover, the strategy to covalently bind a therapeutic metal ion to the polymer ligand successfully imparted the desired antibacterial properties to the prepared CS-based coatings. The results also support the hypothesis that Ga (III)-CS coatings, regardless of the formulation, are not cytotoxic in direct contact with cells, and can act as suitable platforms for initial human osteoblast cell attachment and spreading. Further biological studies, such as investigations into cell growth, cell proliferation, and osteogenic differentiation of stem cells on the ion-loaded coatings, will be performed in order to ascertain the lasting biological advantages of the coatings. In addition, extra functionalization of the coatings with bioactive materials or multiple therapeutic ions will be considered as a possibility to extend the spectrum of beneficial properties that can be delivered by CS based composite coatings.

**Author Contributions:** Conceptualization, M.A.A., Z.H., and A.R.B; methodology, M.A.A. and Z.H.; validation, M.A.A., Z.H., I.D., and A.R.B.; formal analysis, M.A.A., Z.H., I.D., and A.R.B.; investigation, M.A.A. and Z.H.; resources, A.R.B. and I.D.; writing—original draft preparation, Z.H.; writing—review and editing, M.A.A., Z.H, I.D., and A.R.B.; supervision, A.R.B. and I.D.; project administration, A.R.B.; funding acquisition, M.A.A. All authors have read and agreed to the published version of the manuscript

**Funding:** Muhammad Asim Akhtar would like to thank Higher Education Commission (HEC) of Pakistan and the German Academic Exchange Service (DAAD) for granting a scholarship to pursue doctoral studies. The authors also would like to thank the European Virtual Institute on Knowledge-based Multifunctional Materials AISBL (KMM-VIN) (http://www.kmm-vin.eu/fellowships/) research fellowship program for funding.

**Conflicts of Interest:** The authors declare no conflict of interest.

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
