# Peer review of "Electrophoretic Deposition and Characterization of Functional Coatings Based on an Antibacterial Gallium (III)-Chitosan Complex"

_coatings, doi:10.3390/coatings10050483_

Round 1

Reviewer 1 Report

The manuscript entitled “Electrophoretic deposition and characterisation of two functional coatings based on antibacterial gallium 3 (III)-chitosan complex" put forward a potential approach to develop biofunctionalised metallic surfaces through coating. The study is well rounded (considers both mechanical and biological aspects) and uses well established experimental methodologies at its highest scientific rigour. Considering the increasing requirement of biofunctionalisation for implanted devices, the information presented in the manuscript can become invaluable for the wider biomedical/biomaterial research community. It is for these reasons; I recommend that paper to be accepted for publication following some minor revision as listed below.

  1. The paper is generally well written; however, some persistent errors in sentence structure and grammar were observed. Therefore, a further proofreading may improve the overall clarity of the manuscript. For example: line 493: “which is highly advantageous for the design of tailormade CS-based implant coatings”.
  2. Can the authors elaborate the discussion on their observations of the scratch test: especially “However, 420 high concentrations of Ga (III) impair the regular structure of the CS matrix, inducing cracks (Figure 421 1E), which leads to lower critical loads”. The reasons for the observed behaviour are not clear.
  3. Include discussion of the significance of wettability from a biofunctionalization point of view? What is the significance of this parameter from a biomaterial perspective?
  4. The introduction can be improved by contextualising the significance of the work in the wider sense, some suggested literature below.
  • Extra low interstitial titanium based fully porous morphological bone scaffolds manufactured using selective laser melting. Journal of the Mechanical Behavior of Biomedical Materials, Volume 95, July 2019, Pages 1-12.
  • Mechanical performance of highly permeable laser melted Ti6Al4V bone scaffolds. Journal of the Mechanical Behavior of Biomedical Materials
  • Volume 102, February 2020, 103517.
  • Effect of nano-Al2O3 addition on the microstructure and erosion wear of HVOF sprayed NiCrSiB coatings. Materials Research Express 7 (1), 015006.
  • Parametric optimisation of high-velocity oxy-fuel nickel-chromium-silicon-boron and aluminium-oxide coating to improve erosion wear resistance. Materials Research Express 6 (9), 096560.

Author Response

Comment

1. The paper is generally well written; however, some persistent errors in sentence structure and grammar were observed. Therefore, a further proofreading may improve the overall clarity of the manuscript. For example: line 493: “which is highly advantageous for the design of tailormade CS-based implant coatings”.

Our Response:

We acknowledge the reviewer’s suggestion. We have proofread the paper and corrected to the best of our knowledge. Changes are made accordingly and highlighted in the revised manuscript (line 533-534)

Comment

2. Can the authors elaborate the discussion on their observations of the scratch test: especially “However, 420 high concentrations of Ga (III) impair the regular structure of the CS matrix, inducing cracks (Figure 421 1E), which leads to lower critical loads”. The reasons for the observed behaviour are not clear.

Our Response:

We appreciate this remark. The reasons for the observed behavior in the scratch test are discussed in more detail (line 457-459)

Comment

3.Include discussion of the significance of wettability from a biofunctionalization point of view? What is the significance of this parameter from a biomaterial perspective?

Our Response:

We acknowledge the reviewer’s suggestion. The wettability is an important parameter in relation to the biological interaction of the coating. We expanded the discussion about this point. Changes are made accordingly and highlighted in the revised manuscript (line 366- 372)   

Comment

4. The introduction can be improved by contextualising the significance of the work in the wider sense, some suggested literature below.

Our Response:

Thank you for the valuable suggestion. We had expanded the Introduction as suggested. Changes are highlighted in the revised manuscript and some relevant refs. were added (refs. /27-29/) (line 73- 77)

Reviewer 2 Report

Several comments must be resolved:

  1. not clear if the Ga(III) -CS complex was tested before, and the novelty resides on the deposition method and complexity of the work;
  2. the introduction does not provide literature background on the materials used before, based on Ga(III) or chitosan;
  3. A table perhaps will resolve the issue;
  4. what is the Ga(III) solution zeta potential?
  5. did the CS solution alone presented cracks on SEM? what about on the different ratios not presented in the paper?
  6. 89 degrees is not mildly hydrophobic is actually hydrophylic; indeed it changed from just chitosan that is around 78 degrees but not clear what was the point of having this analysis;

Author Response

Reviewer #2:

Comment

1.Not clear if the Ga(III) -CS complex was tested before, and the novelty resides on the deposition method and complexity of the work;

Our Response:

The actual preparation (synthesis) and characterization Ga(III)-CS complex are not part of this paper, this is being prepared for another publication. The present study focuses on the electrophoretic deposition of such Ga (III)-CS complex, which has never been discussed before.

Comment

2.The introduction does not provide literature background on the materials used before, based on Ga(III) or chitosan; A table perhaps will resolve the issue;

Our Response:

3.We acknowledge the reviewer’s suggestion. More relevant literature is included in the revised manuscript and highlighted (line 67- 69). We feel that adding a Table will not fulfil the purpose here.

Comment

4.What is the Ga(III) solution zeta potential?

Our Response

The data is not available. We consider that while the zeta potential of the pure gallium solution would be “nice to have” as additional data, the lack of this specific information does not modify the scientific output and the conclusions of the paper, as EPD has taken place following the standard EPD mechanism of chitosan (explained in Section 3.1 and in reference /32/).

Comment

5.Did the CS solution alone presented cracks on SEM? What about on the different ratios not presented in the paper?

Our Response

Pure CS coatings did not exhibit any cracks on their surface (as shown in Figure 1A and B). Similarly, 1:32, 1:16 and 1:8 CS-Ga (III) coatings were also crack-free. This is mentioned in line 285-287

Comment

6. 89 degrees is not mildly hydrophobic is actually hydrophylic; indeed it changed from just chitosan that is around 78 degrees but not clear what was the point of having this analysis;

Our Response

We acknowledge the reviewer’s remark. The purpose of the analysis and the required corrections are highlighted in the manuscript (line 366-372, line 379-380 and line 382-384)

Reviewer 3 Report

In the introduction, some logic is lacking.

  1. Why the EPD is used in this work among many coating methods? What is the advantage of the technique?
  2. Why the two-step method is applied? Just avoiding the problem seen in the single-step method? Then, what about the three-step method?
  3. Why Ga(III) is selected in this paper? In a previous work done by the author, CS-Cu(II) was tested with the two-step method. This paper's originality is just checking the two-step method with different metal ions?
  4. What is the purpose of this work? Antibacterial coating? Then why the authors tested only gram-negative, not gram-positive? And why cell viability test is employed on MG-63, not other cell types? The author's thoughts and a big picture of this work should be well written in the introduction.

Fig. 1.
The thickness of the deposited layer of CS vs 1:4 CaIII:CS- complex differs. The thickness difference between sample compositions (1:32, 1:16, 1:8, 1:4, etc.) might give an error in the estimation of characteristics of the sample, such as coating stability and cell viability (i.e., figures 6 and 8).

Fig. 6.
The scratch test may be performed to estimate the stability of the coating. In the figure, however, it is hard to discriminate against which condition of the ratio of CS:GaIII is the best. Is quantification available from the image? For example, calculation of the damaged area...

Fig. 8.
The cell viability test showed low cell viability (~60%). Does the author think that this data is okay for an application like tissue implantation?

Author Response

Reviewer #3:

Comment

1.Why the EPD is used in this work among many coating methods? What is the advantage of the technique?

Our Response:

We acknowledge the reviewer’s remark. We thought this had been sufficiently stated in the Introduction. In any case, some additions are included in the revised manuscript (line 73-77)

Comment

2.Why the two-step method is applied? Just avoiding the problem seen in the single-step method? Then, what about the three-step method?

Our Response:

The need to apply a two-step method was indicated in the original paper, it was applied to avoid the disadvantages of the single-step method (clearly mentioned in line 86-87) and to keep the potential simplicity and versatility in processing. A three-step method is not relevant.

Comment

3.Why Ga(III) is selected in this paper? In a previous work done by the author, CS-Cu(II) was tested with the two-step method. This paper's originality is just checking the two-step method with different metal ions?

Our Response:

We acknowledge the reviewer’s remark. The purpose of the work is now described in more detail in the introduction (line 92-102). Other purpose of this work is to present the CS chelation properties with a trivalent ion i.e. Ga(III) in electrophoretic deposited coatings (for the first time).

Comment

4.What is the purpose of this work? Antibacterial coating? Then why the authors tested only gram-negative, not gram-positive? And why cell viability test is employed on MG-63, not other cell types? The author's thoughts and a big picture of this work should be well written in the introduction.

Our Response:

We acknowledge the reviewer’s suggestion. Again, we had thought that the Introduction was clear enough. Certainly the overall purpose of the work is now more clearly stated in the introduction (line 39-44). The purpose of using MG-63 is clarified in lines 214-215.

Comment

Fig. 1.

The thickness of the deposited layer of CS vs 1:4 Ga (III)-CS- complex differs. The thickness difference between sample compositions (1:32, 1:16, 1:8, 1:4, etc.) might give an error in the estimation of characteristics of the sample, such as coating stability and cell viability (i.e., figures 6 and 8).

Our Response:

We acknowledge the reviewer’s suggestion. The thickness of the sample compositions 1:32, 1:16, 1:8 was the same as that of the pure CS coating. However only the thickness of the 1:4 differs from the others (mentioned in line 303-307). In cell studies (figure 8) no statistically significant differences were observed in the cell viability on the different coatings. However, in figure 6 a lower critical load was observed for the 1:4 sample that was due to the presence of cracks. More detail is mentioned in line 457-459.

Comment

Fig. 6.

The scratch test may be performed to estimate the stability of the coating. In the figure, however, it is hard to discriminate against which condition of the ratio of CS:GaIII is the best. Is quantification available from the image? For example, calculation of the damaged area...

Our Response

The scratch test includes usually two stages, cohesive and adhesive. The cohesive stage consists of the onset of microcracking, crack interconnection and transition from microcracking to onset of buckling. Depth and width of the scratch are usually taken into quantitative evaluations of this stage whereas the area being damaged by the buckling can be taken is morphological characteristics in the buckling stage. Sometimes, a well-marked intermediate step consisting of microcracking and microcrack interconnection can be observed (typically for the composite 1:16 (Fig. 6 E,F) and less for 1:8 composite (Fig. 6 G,H)). These phenomena are sometimes taken as material property related to its fracture resistance. In this study the interest has been focused on the cohesive stage of the coatings performance under scratch loading, which is important for the coating mechanical functionality, and for which the critical load is taken as clearly defined parameter. 

We acknowledge the reviewer’s remarks. Certainly, from these images we quantified the critical loads of the coatings, presented in table 3.

Comment

Fig. 8.

The cell viability test showed low cell viability (~60%). Does the author think that this data is okay for an application like tissue implantation?

Our Response

The in vitro cell biology data presented cannot be directly extrapolated to a clinical application. The cell culture study is relevant to assess the potential cytotoxicity of the coatings. It should be taken into account that cells were directly seeded on the surface of the coatings (direct-contact test) so the coatings surface properties might influence the attachment and spreading of the cells. This could be a reason for the lower cell viability on all types of coatings compared to the smooth surface of the control (tissue culture plate). However, if we compare coatings with different concentration of Ga (III) with CS coatings no statistical difference was observed. In vitro cell tests can give a prediction about potential material toxicity. In order to draw general conclusions about the coating toxicity to tissues, further in vivo and clinical studies are required.